Chronological age, biological age, and individual variation in the stress response in the European starling: a follow-up study

Gott Annie 1
http://orcid.org/0000-0002-7484-4447 Andrews Clare 1
Larriva Hormigos Maria 2
Spencer Karen 2
http://orcid.org/0000-0002-0861-0191 Bateson Melissa 1
Nettle Daniel 1 daniel.nettle@ncl.ac.uk
1 Centre for Behaviour and Evolution & Institute of Neuroscience, Newcastle University , Newcastle upon Tyne , UK
2 School of Psychology and Neuroscience, University of St. Andrews , St. Andrews , UK
Bentley George
Electronic publication date: 2018 Oct 23
Publication date: 2018
Volume: 6
Electronic Location ID: e5842
Received 2018 Jul 24; Accepted 2018 Sep 27
Copyright: © 2018 Gott et al.
Copyright year: 2018
Copyright holder: Gott et al.
License: This is an open access article distributed under the terms of the Creative Commons Attribution License, which permits unrestricted use, distribution, reproduction and adaptation in any medium and for any purpose provided that it is properly attributed. For attribution, the original author(s), title, publication source (PeerJ) and either DOI or URL of the article must be cited.
License URL: https://creativecommons.org/licenses/by/4.0/

Keywords: Biological age, Stress response, Corticosterone, Developmental programming, Starlings, Telomeres

Funding: Biotechnology and Biological Sciences Research Council (BBSRC) under grants BB/J016446/1 and BB/J016292/1 European Research Council (ERC) under the European Union’s Horizon 2020 research and innovation programme AdG 666669 (COMSTAR) This research was funded by the Biotechnology and Biological Sciences Research Council (BBSRC) under grants BB/J016446/1 and BB/J016292/1; a doctoral training studentship to Annie Gott; and a David Phillips fellowship to Karen Spencer. The project has also received funding from the European Research Council (ERC) under the European Union’s Horizon 2020 research and innovation programme (grant agreement no. AdG 666669 (COMSTAR)). The funders had no role in study design, data collection and analysis, decision to publish, or preparation of the manuscript.

==============================
The strength of the avian stress response declines with age. A recently published study of European starlings (Sturnus vulgaris) found that a marker of biological age predicted the strength of the stress response even in individuals of the same chronological age. Specifically, birds that had experienced greater developmental telomere attrition (DTA) showed a lower peak corticosterone (CORT) response to an acute stressor, and more rapid recovery of CORT levels towards baseline. Here, we performed a follow-up study using the same capture-handling-restraint stressor in a separate cohort of starlings that had been subjected to a developmental manipulation of food availability and begging effort. We measured the CORT response at two different age points (4 and 18 months). Our data suggest a decline in the strength of the CORT response with chronological age: peak CORT was lower at the second age point, and there was relatively more reduction in CORT between 15 and 30 min. Individual consistency between the two age points was low, but there were modest familial effects on baseline and peak CORT. The manipulation of begging effort affected the stress response (specifically, the reduction in CORT between 15 and 30 min) in an age-dependent manner. However, we did not replicate the associations with DTA observed in the earlier study. We meta-analysed the data from the present and the earlier study combined, and found some support for the conclusions of the earlier paper.

Introduction

The hypothalamic-pituitary-adrenal (HPA) axis is a highly conserved, integrated system in vertebrates that functions to prioritise immediate survival over non-essential activities in the face of acute threats. In birds, the strength of the HPA response generally declines with chronological age (Heidinger, Nisbet & Ketterson, 2006; Heidinger et al., 2010; Wilcoxen et al., 2011; Elliott et al., 2014; Lendvai, Giraudeau & Bo, 2015; López-Jiménez et al., 2017), possibly reflecting adaptive shifts in behavioural allocation as expected future lifespan reduces. However, individuals do not all age at the same rate: an individual’s biological age can be either older or younger than their chronological age (Belsky et al., 2015). Biological age is by definition a better predictor of future lifespan than chronological age is. Hence, we should expect markers of individual biological age to explain variation in the strength of the stress response that cannot be explained by chronological age alone. A possible reason that early-life conditions have often been observed to influence the functioning of the adult stress response may be that early-life conditions can speed up or slow down the biological ageing process, and consequently advance or retard age-related shifts in the functioning of the stress response system.

One potential marker of biological age is the attrition of telomeres, the DNA caps at the ends of linear chromosomes (Bize et al., 2009; Bauer et al., 2018). Short telomere length has been shown to predict shorter subsequent lifespan in a range of avian (Bize et al., 2009; Heidinger et al., 2012; Barrett et al., 2013; Salmón et al., 2017; Wilbourn et al., 2018) as well as non-avian (Boonekamp et al., 2013; Wilbourn et al., 2018) species. The rate of telomere attrition is much higher during the developmental period than in adulthood, and has been shown to be accelerated by early-life adversity (Boonekamp et al., 2014; Nettle et al., 2015, 2017). Thus, the extent of telomere shortening can potentially be used as a proxy for individual differences in biological age.

In a recent study of two cohorts of adult European starlings (Sturnus vulgaris, n = 20 and n = 31), Andrews et al. (2017) showed that developmental telomere attrition (DTA; the extent of shortening of erythrocyte telomeres over the course of development) explained variation in individuals’ stress responses. Specifically, individuals that were biologically older by this measure (all individuals were approximately the same chronological age) showed a lower peak level of corticosterone (CORT) in response to an acute capture-handling-restraint stressor, and also showed stronger recovery of CORT levels towards baseline between 15 and 30 min after the onset of the stressor. DTA was unrelated to baseline CORT. The birds studied by Andrews et al. (2017) had been subjected to experimental manipulations of early-life conditions: a manipulation of brood size for one cohort (Nettle et al., 2013), and of the focal individual’s size relative to its competitors in the other cohort (Nettle et al., 2015). These manipulations affected DTA, with the more adverse treatment (having more or larger competitors respectively for the two cohorts) leading to greater telomere shortening in early life. However, it was DTA, rather than the early-life treatments themselves, that significantly predicted stress response. This suggests that DTA captures both the adversity due to the experimental manipulation, and other sources of adversity, and also incorporates the fact that individuals are differentially affected by the same external conditions. Thus, DTA is a better marker of biological ageing than any of the individual environmental or genetic factors that may influence it. The cohorts of birds were made up of quartets of siblings, which allowed the researchers to examine potential familial effects. They found evidence consistent with modest familial effects on baseline and peak CORT, but not on the change in CORT between 15 and 30 min after the onset of the stressor.

In light of recent focus on reproducibility in bioscience (Fidler et al., 2017), the results of Andrews et al. (2017) require replication. Moreover, the CORT response was only measured at one age point (around one year of age on average). Thus, it is not clear whether associations between DTA and stress response appear before, or persist after, this stage of life. Longitudinal studies in birds have found varying results with respect to whether individuals are consistent in CORT measures from one year to the next (Cockrem & Silverin, 2002; Angelier et al., 2010; Ouyang, Hau & Bonier, 2011; Rensel & Schoech, 2011; Baugh et al., 2014; Lendvai, Giraudeau & Bo, 2015). Where individual consistency is found, it is generally in the CORT response to an acute stressor rather than baseline CORT (Cockrem & Silverin, 2002; Rensel & Schoech, 2011). Thus, it is important to establish whether there is individual consistency over time in the CORT parameters studied in Andrews et al. (2017), as well as whether the associations of these parameters with DTA are robust.

Here, we report a follow-up study conducted with a different cohort of birds. The quartets of siblings in the current cohort were hand-reared according to a two-by-two factorial design that varied early food amount and begging effort. The rationale for this design was to attempt to isolate the independent effects of these two factors, both of which vary with brood size in nature. Previous studies of early-life adversity in altricial birds have found that developing in a large brood has long-term phenotypic consequences (Nettle et al., 2013; Boonekamp et al., 2014), but have not been able to establish exactly why. We have reported the effects of the developmental treatments on DTA in these birds elsewhere (Nettle et al., 2017). Briefly, a restricted food amount, and a higher begging effort, both significantly accelerated DTA in these birds, and did so additively. Thus, the two developmental treatments can each be seen as accelerating biological ageing, as proxied by DTA.

In the present experiments, we measured the CORT response to an acute capture-restraint-handling stressor using the same protocol as Andrews et al. (2017), but did so twice, when the birds were juveniles of approximately 4 months of age (age point 1), and adults of approximately18 months of age (age point 2). Our main aim was to replicate the key findings of Andrews et al. (2017), namely that DTA is associated with peak CORT and change in CORT between 15 and 30 min, but not baseline CORT; that DTA is a better predictor of CORT parameters than the early-life conditions to which birds have been exposed; and that there are modest familial effects on baseline and peak CORT. In addition, our study, through having two age points, gave us the opportunity to ask a number of additional questions. First, it may be that the association between DTA and the strength of the stress response develops with chronological age. Our age point 2 was most similar to the ages of the birds in the study by Andrews et al. (2017). Thus, it is possible that the association between DTA and CORT variables will be observable at age point 2, but not yet in evidence at age point 1. Second, in their interpretation of their findings, Andrews et al. (2017) assumed that the strength of the stress response declines with chronological age in the starling, basing this assumption on published literature from other bird species. In our study, we were able to directly test this assumption within individuals. Third, our repeated-measure design allowed us to characterise individual consistency in CORT variables in the starling.

Our study suffered from a large limitation in respect of the analyses of within-individual change with chronological age, and individual consistency. Unfortunately, the second age point sample was not planned at the time the first was carried out. We therefore performed the laboratory assays on the blood samples from the first age point separately to the second, and with a small variation in the laboratory protocol. Although common standard samples were run on both occasions, this does limit the robustness of inferences we can make about within-individual change and consistency, since age point and laboratory batch are completely confounded. The findings in respect of within-individual change with age, and individual consistency, should therefore be interpreted with caution.

Materials and Methods

Study subjects and husbandry

Study subjects were from a cohort of 32 starlings (16 male) hatched in 2014 (hence the ‘2014 cohort’) described in detail elsewhere (Nettle et al., 2017). Briefly, quartets of natural nest-mates were taken from the wild on day 5 post-hatch and hand-reared. One sibling was assigned to each combination of food Amount (‘Plenty’ vs. ‘Lean’) and begging Effort (‘Easy’ vs. ‘Hard’), thus creating four experimental groups of eight birds per group. The Amount manipulation was achieved by feeding each Plenty group to satiation on each feed and measuring the quantity consumed, then restricting the intake of the corresponding Lean group to 73% of this quantity. The effort manipulation was achieved by interspersing, for the Hard groups, each true feed with another nest visit of similar duration where the nestlings were stimulated to beg but no food was delivered. The experimental treatments lasted until day 15, after which all birds were hand-fed ad libitum until independence. Birds were subsequently kept in mixed-treatment flocks in indoor aviaries (215 × 340 × 220 cm; 18 °C, 40% humidity), with ad libitum access to food and water. Birds were maintained in non-breeding conditions by a constant 15 h light: 9 h dark cycle.

Our study adhered to the ASAB/ABS Guidelines for the Use of Animals in Research, and was approved by the Animal Welfare and Ethics Review Board at Newcastle University. It was completed under UK Home Office project licence number PPL 70/8089.

Developmental telomere attrition

Telomere length was measured in erythrocyte DNA from blood samples taken on day 5 and day 56 post-hatch using a real-time PCR amplification method (for details see Nettle et al., 2017). We used the two T/S ratio values to gain a single-number summary of telomere shortening (henceforth ΔTL). ΔTL was corrected for the regression to the mean using the method of Verhulst et al. (2013). Consequently, 0 represents the average amount of change for the cohort, and a negative number more dramatic shortening. Similar results are obtained using the raw difference in T/S ratios instead.

Blood sampling and corticosterone assays

Birds were aged 127–134 days at age point 1, and 584–601 days at age point 2. For the period of sampling, birds were individually caged (75 × 45 × 45 cm) whilst maintaining full acoustic and visual contact with others. The minimum time in individual cages prior to sampling was 3 days. All birds had access to two wooden perches, two drinking bottles, a water bath (removed approximately 4 h prior to sampling) and a bowl containing ad libitum food. The birds were maintained in environmental conditions identical to the free-flight aviaries. Habituation to the cages occurred over a minimum of three nights prior to blood sampling.

At a set time during the afternoon (the period of minimal diurnal CORT variation, Romero & Remage-Healey, 2000), lights were extinguished and two birds caught from their cages and transferred immediately to an adjacent procedure room. Approximately 120 μl of a baseline blood sample was collected within 3 min of the lights being extinguished (age 1: mean time to baseline sample: 94.9 ± 23.6 s; age 2: 108.4 ± 29.7 s). One bird was removed from analysis at age 1 as the time to baseline sample was at the 180 s limit and visual analysis of the radioimmunoassay data confirmed a very high level of baseline CORT. Bleeding was stemmed using cotton wool and birds were placed in drawstring cloth bags. Further samples were taken at 15 and 30 min after the initial disturbance. Blood was taken by puncture of an alar or metatarsal vein, and collection by heparinised micro-capillary tubes. After the final sample, birds were weighed and returned to the cage under observation. Each experimental room was disturbed for sampling only once per day, and no-one entered the room for at least 2 h prior to a sample being taken.

Blood samples were centrifuged (10 min at 3,000 rpm) to separate plasma from erythrocytes and stored at −80 °C until radioimmunoassay analysis. Samples from age point 1 and age point 2 were analysed at different times. Batch 1 (age point 1) consisted of two separate assays, and batch 2 (age point 2) consisted of four assays. The same protocol was followed for both batches, and two standard chicken plasma samples (P3266-1ML; Sigma Aldrich, St. Louis, MO, USA) were run in all assays in both batches. All samples were run in duplicate within the same assay. Freezer storage times were approximately 120 days (batch 1) and 30 days (batch 2).

Corticosterone levels in plasma extracts were quantified using a radioimmunoassay previously validated in European starlings (Buchanan et al., 2003). CORT concentrations were measured after extraction of up to 35 μl aliquots of plasma in one ml diethyl ether (24004-2.5L-M; Sigma Aldrich, St. Louis, MO, USA) by a Dextran 70-coated charcoal radioimmunoassay method. The anti-CORT serum was ABIN880 (Antibodies Online, London, UK) for batch 1 and 07120016 (MP Biomedical, Loughborough, UK) for batch 2. Extraction efficiencies per sample were estimated at 61–100% (mean 97.8%) for batch 1, and 77–100% (mean 96.1%) for batch 2. Final CORT concentration values were corrected accordingly.

The average intra-assay coefficient of variation was 8.7% for batch 1 and 13.5% for batch 2. The inter-assay coefficient of variation (collapsing across batches) was 21.9%. The batch 2 average concentration values for the two control samples were 99.84% and 78.15% of the batch 1 values, respectively.

Data analysis

As in Andrews et al. (2017), the dynamics of the stress response were characterised by three dependent variables: baseline CORT (the first sample value); peak CORT (higher of second and third sample values); and ΔCORT (the change in CORT value between 15 and 30 min samples, with a negative number indicating a reduction from 15 to 30 min).

Our main data-analytic approach was to fit linear mixed models of the data from age points 1 and 2 combined, for each of the three outcome variables (baseline CORT, peak CORT, and ΔCORT), in turn. All models contained random effects of bird, nested within natal family. To examine individual consistency and familial resemblance, we fitted initial models with no fixed predictors. The proportion of variation explained by the random effect of bird in these initial models is a measure of individual consistency. An alternative approach to estimating individual consistency is to calculate the intra-class correlation coefficient between the age point 1 and age point 2 measures. We also did this, and report these in the ‘Results’ section. The extent of familial resemblance can be estimated from the initial models using the proportion of variation explained by the random effect of natal family.

For the main objectives, we added ΔTL, age point, and the interaction between ΔTL and age point to the initial models as fixed predictors. These models thus test for within-individual changes with chronological age; associations between ΔTL and CORT variables after controlling for age point; as well as differences between the age points in terms of the association between ΔTL and the outcome variable. As in Andrews et al. (2017), we included baseline CORT as a covariate in the model where the outcome variable was peak CORT; and CORT at 15 min as a covariate in the model where the outcome variable was ΔCORT. We did not include additional covariates (sex, body weight, time elapsed before baseline sample) that we might otherwise have considered, since these were not included in the analyses of the previous paper. We can report that including these does not substantively alter the results presented here.

There are two drawbacks to our modelling approach: first, it does not involve exactly the same statistical models as Andrews et al. (2017), since their data included only one age point; and second, it combines the data from the two laboratory batches, which could be misleading if the assays from the two batches are not validly comparable. For the replication of the associations with ΔTL, we therefore also ran additional analyses treating the age point 1 and age point 2 datasets as completely separate, and for each one in turn, fitting the same statistical models as Andrews et al. (2017).

In order to establish the current balance of evidence on associations between ΔTL and CORT variables, we performed fixed-effects meta-analyses on all three cohorts of birds (the two from the previous paper plus the present one; the cohorts from the earlier paper are referred to as 2012 and 2013). Separate meta-analyses were performed using age point 1 to represent the 2014 cohort, and using age point 2. Parameter estimates were recalculated as standardised βs for this purpose.

To investigate effects of developmental treatments, we fitted models of each CORT variable with Amount, Effort, age point and all their two-way interactions as fixed predictors. Again, baseline CORT was included as a covariate for the model of peak CORT, and CORT at 15 min for the model of ΔCORT.

All analyses were performed in R Version 3.5.0 (R Core Development Team, 2018), using the contributed packages ‘irr’ (Gamer et al., 2012) for intra-class correlation coefficients, ‘afex’ (Singmann et al., 2018) for linear mixed models, and ‘metafor’ (Viechtbauer, 2010) for meta-analysis. Categorical variables (the developmental treatments) are automatically contrast-coded by the ‘afex’ package. Type-III significance tests for linear mixed models were by likelihood ratio test (LRT) with a significance threshold of 0.05, and hence parameter estimation was by maximum likelihood.

We also created a simple R simulation tool to simulate the chances of finding a significant (p < 0.05) association between two variables in small samples, for a given true strength of association. This tool generates 10,000 samples of a specified size from datasets where the true strength of association is as specified, and tabulates how many of them find a significant effect, how many a non-significant effect but in the predicted direction, and how many an effect in the opposite direction. This tool thus simulates the power of either our age point 1 or age point 2 data considered separately to replicate the significant associations observed in Andrews et al. (2017).

Raw data and R scripts are available for download from: https://doi.org/10.5281/zenodo.1317793.

Results

Final sample sizes

Of the original 32 birds in the cohort, we failed to obtain CORT variables at age point 1 for one bird, and two birds died between age point 1 and age point 2. Thus, maximum samples sizes for CORT variables were 31 for age point 1 and 30 for age point 2, with 29 birds having data at both time points. In addition, due to telomere assay failures (see Nettle et al., 2017), ΔTL was unavailable for five birds, including one of the birds that died between age point 1 and age point 2. Thus, for analyses involving ΔTL, the maximum sample size is 26 at age point 1 and 26 at age point 2, with 25 birds having data at both time points.

Descriptive statistics

Descriptive statistics for CORT values at all time points and both age points are shown in Table 1. At both age points, the baseline blood sample gave the lowest CORT value for all birds (Fig. 1). Some birds’ CORT values were highest at 15 min and then declined by 30 min, giving ΔCORT values less than 0 (16/31 at age point 1, 16/30 at age point 2). The remainder continued to show an increase between 15 and 30 min.

Table 1 Descriptive statistics (mean ± SD) for CORT variables (ng/ml) at age point 1 and age point 2.

Variable	Age 1 (127–134 days)	Age 2 (584–601 days)	
Baseline CORT	2.21 ± 1.99	2.76 ± 1.53	
CORT 15 min	19.58 ± 8.89	13.89 ± 6.23	
CORT 30 min	18.44 ± 6.41	13.25 ± 5.27	
Peak CORT	21.66 ± 8.50	15.48 ± 5.77	
ΔCORT	−1.14 ± 7.00	−0.64 ± 5.01	
Note:

Peak CORT represents the higher of CORT 15 min and CORT 30 min; ΔCORT represents CORT 30 min minus CORT 15 min.

Figure 1 CORT values for individual birds at baseline, 15 min after onset of stressor, and 30 min after onset of stressor, at the two age points.

Individual consistency and familial effects

Our initial models contained only the random effects of bird nested within natal family. For baseline CORT and peak CORT, bird explained <1% of the variation in the initial models. This suggests negligible individual consistency in these CORT variables. This was confirmed by calculating intra-class correlation coefficients between age point 1 and age point 2 (baseline CORT 0.13, 95% CI [−0.24 to 0.46]; peak CORT −0.03, 95% CI [−0.38 to 0.33]). For ΔCORT, bird explained 35% of the variation, suggesting some degree of individual consistency. This was confirmed by the intra-class correlation coefficient, which was higher than for the other two CORT variables, though its 95% confidence interval still included zero (0.32, 95% CI [−0.04 to 0.61]).

Natal nest explained 33% of the variation in baseline CORT; 17% of the variation in peak CORT; and <1% of the variation in ΔCORT (Fig. 2).

Figure 2 Familial and individual components of variation in CORT variables.

Bars represent the proportion of variation explained by natal nest and bird, for the two cohorts of birds studied by Andrews et al. (2017; cohorts referred to as 2012 and 2013), and the cohort studied in the present paper (2014).

Effects of chronological age and ΔTL

Our main models included fixed effects of age point, ΔTL, and their interaction. A complete replication of Andrews et al. (2017) would find significant main effects of ΔTL on peak CORT and ΔCORT but not baseline CORT; a replication at one but not the other age point would find a significant interaction between age point and ΔTL for peak CORT and ΔCORT but not baseline CORT; and a decline in stress response with chronological age would produce a significant main effect of age point on peak CORT and ΔCORT.

The models are summarised in Table 2. For baseline CORT, neither age point, ΔTL, nor their interaction were significant predictors. For peak CORT, there was a significant main effect of age point, with lower peak CORT at age point 2 (estimated marginal means: age point 1, 21.57, 95% CI [17.76–25.38]; age point 2, 15.74, 95% CI [11.94–19.53]). The main effect of ΔTL and the interaction between ΔTL and age point were not significant. For ΔCORT, there was a significant effect of CORT at 15 min, which was expected since birds with higher CORT at 15 min tend to reduce more by 30 min. There was a significant effect of age point, with birds at age point 2 returning more towards baseline between 15 and 30 min (i.e. having a more negative ΔCORT value; estimated marginal means: age point 1, 0.27, 95% CI [−2.37 to 2.90]; age point 2, −2.41, −5.01 to 0.19). The main effect of ΔTL and its interaction with age point were not significant.

Table 2 Summaries of statistical models testing for effects of age point and developmental telomere attrition (DTA) on stress response (CORT) variables.

Outcome variable	Fixed predictors	B	s.e. (B)	LRT	p-value	
Baseline CORT	Age point	0.51	0.40	1.65	0.20	
DTA	−0.66	1.25	0.28	0.60	
Age point × DTA	2.54	1.66	2.27	0.13	
Peak CORT	Baseline CORT	0.55	0.55	0.92	0.34	
Age point	−5.82	1.64	11.18	<0.001*	
DTA	3.02	5.09	0.35	0.55	
Age point × DTA	−4.37	6.93	0.39	0.53	
ΔCORT	CORT 15 min	−0.61	0.08	33.10	<0.001*	
Age point	−2.69	1.12	5.26	0.02*	
DTA	−0.03	3.24	0.00	0.99	
Age point × DTA	2.27	4.31	0.28	0.60	
Notes:

All models contain random effects of bird nested within family.

LRT, likelihood ratio test.

* p < 0.05.

We additionally performed analyses on the two age point datasets separately, fitting to each one exactly the same statistical models used in the study by Andrews et al. (2017). These models are summarised in Table S1; they lead to the same conclusion that ΔTL does not significantly predict any CORT variable in the present cohort of birds.

Meta-analysis

The lack of significant effects involving ΔTL in the present study contrasts with the findings of Andrews et al. (2017). To evaluate the balance of evidence, we performed fixed-effects meta-analyses combining the data from the present study with the data from the earlier paper (Fig. 3). As the figure shows, the summary association between ΔTL and baseline CORT was not significantly different from zero, regardless of whether the age point 1 or age point 2 results were used to represent the 2014 cohort. The association between ΔTL and peak CORT after controlling for baseline CORT was significantly positive (that is, more telomere loss associated with a lower peak CORT concentration) if the age point 1 data were used to represent the 2014 cohort (β = 0.30, 95% CI [0.01–0.41], p = 0.04), but not significantly different from zero if the age point 2 data were used (β = 0.16, 95% CI [−0.04 to 0.36], p = 0.12). The summary association between ΔTL and ΔCORT after controlling for CORT at 15 min was significantly positive (that is, more telomere loss, more CORT return towards baseline between 15 and 30 min after onset of stressor). This was true using either the age point 1 (β = 0.30, 95% CI [0.11–0.49], p < 0.01) or age point 2 data (β = 0.30, 95% CI [0.11–0.49], p < 0.01).

Figure 3 Forest plot of associations between developmental telomere change and CORT variables in the present cohort of birds (2014) and the two cohorts described previously (2012 and 2013).

The points and whiskers show standardised parameter estimates and their 95% confidence intervals. The lozenges show summary effects from meta-analytically combining the three datasets (upper lozenges use the age 1 point data for the present cohort, lower lozenges use the age point 2 data).

Effects of developmental treatment

Table 3 summarises the models using the two developmental treatments (Amount and Effort), plus age point and their interactions, as the predictors. There were no significant main effects or interactions involving the developmental treatments for baseline CORT or peak CORT. For ΔCORT, there was a significant interaction between age point and the Effort treatment. Birds from the Hard groups showed more return of CORT towards baseline between 15 and 30 min (i.e. had more negative values of ΔCORT) than birds from the Easy groups at age point 1. However, at age point 2, the ΔCORT of the birds from the Hard groups had scarcely changed on average, whereas the Easy groups had showed substantially reduced ΔCORT with age (Fig. 4).

Table 3 Summaries of statistical models testing for effects of developmental treatments on stress response (CORT) variables.

Outcome	Fixed predictors	B	s.e. (B)	LRT	p-value	
Baseline CORT	Age point	0.48	0.38	1.61	0.20	
Amount	0.03	0.26	0.02	0.90	
Effort	−0.06	0.26	0.05	0.81	
Amount × Effort	0.11	0.19	0.33	0.57	
Amount × Age point	−0.13	0.37	0.11	0.74	
Effort × Age point	−0.02	0.38	0.00	0.97	
Peak CORT	Baseline CORT	1.06	0.51	4.00	0.05*	
Age point	−6.69	1.59	15.43	<0.001*	
Amount	1.40	1.10	1.62	0.20	
Effort	1.17	1.10	1.13	0.29	
Amount × Effort	−0.41	0.78	0.27	0.60	
Amount × Age point	−1.34	1.57	0.73	0.39	
Effort × Age point	−2.18	1.57	1.90	0.17	
ΔCORT	CORT 15 min	−0.58	0.07	38.71	<0.001*	
Age point	−2.77	1.03	6.58	0.01*	
Amount	−0.53	0.69	0.59	0.44	
Effort	1.37	0.67	3.99	0.05*	
Amount × Effort	0.24	0.48	0.24	0.62	
Amount × Age point	0.50	0.96	0.26	0.61	
Effort × Age point	−2.55	0.95	6.68	0.01*	
Notes:

All models contain random effects of bird nested within natal family.

LRT, likelihood ratio test.

* p < 0.05.

Figure 4 Estimated marginal means of ΔCORT after controlling for CORT at 15 min, by levels of the Effort developmental treatment, and age point.

Error bars represent plus/minus one standard error.

Power simulation

To set our failure to replicate the significant patterns involving ΔTL reported in the previous paper into context, we simulated 10,000 samples of 27 individuals from populations where the ‘true’ association between ΔTL and peak CORT was 0.28, which is the pooled estimate from the two cohorts of birds reported in Andrews et al. (2017). Of these 10,000 samples, 30% produced ‘significant’ estimates of association with p < 0.05; 63% produced non-significant associations but with an estimate in the positive direction; and 7% produced estimates of association in the other direction. We repeated the same exercise for the estimate of association between ΔTL and ΔCORT (0.43). This produced ‘significant’ associations 64% of the time, non-significant associations in the same direction 35% of the time, and estimated associations in the opposite direction 1% of the time.

Discussion

The main aim of our experiment was to replicate the findings of Andrews et al.’s (2017) study in a different cohort of birds measured at two time points. We did not replicate the main patterns observed in the previous paper concerning DTA. The change in erythrocyte telomere length over development did not significantly predict any of the CORT parameters in the present cohort, in the whole dataset, or at either age point considered separately. Our failure to clearly replicate the patterns of the previous study does not support the conclusions of that paper. However, we should not necessarily infer that those conclusions were spurious, either. Meta-analysis of all the extant evidence supports a moderate association between DTA and ΔCORT (β = 0.30), with starlings that have experienced more DTA showing more rapid recovery of CORT towards baseline; and possibly between DTA and peak CORT, with starlings that have experienced greater DTA having a lower peak. The former conclusion is supported whether age point 1 or age point 2 is used to represent the present 2014 cohort of birds. The latter conclusion is only supported if the age point 1 data are used. Age point 2 (584–601 days) was more similar to the age at which the birds’ stress responses were measured in the previous study (2012 cohort, 208–432 days; 2013 cohort, 428–456 days). Our simulations showed that if associations between DTA and CORT variables do in fact exist, and have the strength estimated in Andrews et al. (2017), then we should not expect them to be statistically ‘significant’ (i.e. have p < 0.05) in every small-n sample considered individually. For example, if the estimated parameters in Andrews et al. (2017) are correct, then we should only expect a ‘significant’ finding about one experiment in three for peak CORT, and two in three for ΔCORT. (Our two age points constitute two separate experiments in the context of this statement).

We acknowledge the low power of our experiment for detecting associations between DTA and CORT parameters. However, there are strong logistical constraints involved in capturing, keeping, in our case hand-rearing, and sampling live wild animals. For example, to detect an association of β = 0.30 with the conventional 80% power requires over 80 birds, which is beyond the population size of our starling breeding colony or our ability to hand-rear nestlings under precisely controlled conditions. This means that modest sample sizes are difficult to avoid, especially in the early stages of exploration of certain questions. Given that we cannot rear very large cohorts of birds, we have to turn to sequential replication and cumulative meta-analysis, rather than individual-experiment p-values, as a way of ensuring robustness of conclusions.

A further limitation of both this study and Andrews et al. (2017) is that ‘biological age’ is proxied by a single marker, DTA. A better approach for capturing the biological age construct is to use a whole panel of biomarkers, each of which has been shown to predict subsequent lifespan (Levine, 2013; Belsky et al., 2015). Such a panel predicts morbidity and other phenotypic outcomes better than any single constituent biomarker, each of which might have quite weak predictive power. Thus, a stronger test of the hypothesis that individual differences in stress response among birds of the same chronological age are explained by differences in their biological ages would use a panel of multiple biomarkers of biological age, rather than the single one used here.

The three CORT parameters were not individually consistent across the two age points (only ΔCORT showed any suggestion of individual consistency across the two age points, and this was still fairly low). This must be interpreted cautiously given that the samples from the two age points were run in separate laboratory batches. It does however concur with some previous findings in the avian literature that individual consistency can be low (Ouyang, Hau & Bonier, 2011; Baugh et al., 2014; Lendvai, Giraudeau & Bo, 2015); and if consistency is found, is in the response to acute stress rather than baseline CORT (Cockrem & Silverin, 2002; Rensel & Schoech, 2011).

We found some evidence of familial effects. Specifically, natal nest explained a modest but non-zero proportion of the variation in baseline CORT and peak CORT, but a negligible amount of the variation in ΔCORT. The familial effect findings were extremely similar to those of Andrews et al. (2017). Since our birds were removed from the wild on day 5 post-hatching, and siblings were thereafter assigned to different developmental treatment groups, these familial effects must represent either genetics, maternal effects, or very early developmental influences.

We found significant effects of chronological age on both peak CORT (lower at age point 2) and ΔCORT (more return towards baseline at age point 2). Our design was more powerful for detecting these than it was for detecting associations with DTA, as the critical comparisons were within subjects. The findings should be interpreted cautiously, as chronological age was completely confounded with laboratory batch. Nonetheless, they may at least partly represent within-individual biological change with age. There were common control samples run in both batches; the average concentration for these standards at age point 2 was 99.84% and 78.15% of their age point 1 values, whereas peak CORT was 71.47% of its average age 1 value. Decline in the strength of the CORT response (but not baseline CORT) with chronological age is a very widely observed pattern in birds (Heidinger, Nisbet & Ketterson, 2006; Heidinger et al., 2010; Wilcoxen et al., 2011; Elliott et al., 2014; Lendvai, Giraudeau & Bo, 2015; López-Jiménez et al., 2017). Indeed, a chronological age-related decline in CORT response is one of the key assumptions on which Andrews et al. (2017) based their hypotheses, but which they did not directly test. Thus, we have provided some validating evidence for this part of their argument. The differences between age point 1 and age point 2 were strikingly large, and set the associations between DTA and CORT parameters in context. They suggest that any DTA-related differences in stress response amongst birds of the same age are small relative to the within-bird change as a bird develops from a juvenile into an adult.

We found some evidence that one of the developmental manipulations to which the birds were subjected may have affected their stress responses, in an age-dependent manner. Specifically, age point and begging Effort interacted to predict ΔCORT. At the first age point, birds from the Hard begging effort groups had more negative values of ΔCORT than those from the Easy groups. This is consistent with the idea that increased begging effort accelerates biological ageing, since ΔCORT values become more negative with chronological age. It is also consilient with the finding that Hard begging effort accelerated telomere shortening in early life in these birds (Nettle et al., 2017). However, at the second age point, the average ΔCORT had greatly reduced for the Easy birds, who showed the general age-related trend, whilst for the Hard birds it had remained almost unchanged compared to age point 1. Thus, the pattern cannot simply be described as begging effort increasing the rate of ageing of the stress response. Rather, increased begging effort produces in a juvenile bird the stress response that would be expected of an adult; and then that stress response remains the same into adulthood, whilst the stress responses of other birds are changing substantially. Thus, the shape of age-related pattern appears to be different for the Hard birds than for the Easy birds. We note that these developmental treatment findings were not predicted a priori; thus, we view these treatment-related findings as exploratory. However, other studies of these same birds have also found that the Hard Effort treatment leaves a lasting phenotypic legacy: Dunn et al. (2018) showed that the Hard birds maintain a lower body mass than the Easy birds through adulthood, and correspondingly employ different foraging strategies.

In Andrews et al. (2017), a major finding was that developmental conditions did not significantly predict stress parameters, whereas DTA did, the converse of what we see here. Given that developmental treatments were significantly related to DTA in all the cohorts of birds that we have studied, the fact that significant predictor of the strength of the stress response was sometimes DTA (Andrews et al., 2017), and sometimes the developmental treatment itself (here), may relate to limited statistical power, as discussed above.

Conclusions

In conclusion, we did not confirm any significant associations between biological age, as proxied by DTA, and the strength of the stress response, as measured by CORT response to an acute capture-restraint-handling stressor, in a cohort of hand-reared European starlings. However, the overall evidence is still consistent with biological age being associated with aspects of the stress response in starlings. Moreover, our data suggest there is a substantial decline in the strength of the stress response with chronological age in the starling, as the biological age hypothesis requires. They also suggest that early-life begging effort may affect the strength of the stress response in an age-dependent manner; and confirm earlier observations of familial effects on baseline and peak CORT.

Supplemental Information

Supplemental Information 1 Summaries of statistical models testing for effects of developmental telomere attrition (DTA) on stress response (CORT) variables, in the age point 1 and age point 2 data treated separately.

All models contain random effects of natal family. LRT: Likelihood ratio test; * p < 0.05. Age point 1: 127–134 days; age point 2 584–601 days.

Click here for additional data file.

We thank Pat Monaghan and Sophie Reichert, who collaborated with us on the telomere attrition of the birds studied in this paper; and Tom Bedford and Michelle Waddle, who assisted with us in the care of the birds.

Additional Information and Declarations

Competing Interests

Author Contributions

Animal Ethics

Data Availability

The authors declare that they have no competing interests.

Annie Gott conceived and designed the experiments, performed the experiments, analysed the data, prepared figures and/or tables, authored or reviewed drafts of the paper, approved the final draft.

Clare Andrews conceived and designed the experiments, performed the experiments, analysed the data, prepared figures and/or tables, authored or reviewed drafts of the paper, approved the final draft.

Maria Larriva Hormigos conceived and designed the experiments, authored or reviewed drafts of the paper, approved the final draft, hormonal analysis.

Karen Spencer conceived and designed the experiments, analysed the data, contributed reagents/materials/analysis tools, authored or reviewed drafts of the paper, approved the final draft, hormonal analysis.

Melissa Bateson conceived and designed the experiments, performed the experiments, contributed reagents/materials/analysis tools, authored or reviewed drafts of the paper, approved the final draft.

Daniel Nettle conceived and designed the experiments, performed the experiments, analysed the data, prepared figures and/or tables, authored or reviewed drafts of the paper, approved the final draft.

The following information was supplied relating to ethical approvals (i.e. approving body and any reference numbers):

Our study adhered to the ASAB/ABS Guidelines for the Use of Animals in Research, and was approved by the Animal Welfare and Ethics Review Board at Newcastle University. It was completed under UK Home Office project license number PPL 70/8089.

The following information was supplied regarding data availability:

Gott, Annie, Andrews, Clare, Larriva, Maria, Spencer, Karen, Bateson, Melissa, & Nettle, Daniel. (2018). Data archive for Gott et al. ‘Chronological age, biological age, and individual variation in the stress response in the European starling: A follow-up study’ [Data set]. Zenodo. http://doi.org/10.5281/zenodo.1408580.

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
