# Peer review of "Chronological age, biological age, and individual variation in the stress response in the European starling: a follow-up study"

_PeerJ, doi:10.7717/peerj.5842_

## Round 0.1 · original submission · Major Revisions

Please pay close attention to the reviewer comments. I look forward to receiving the revised version.

·

Basic reporting

The manuscript, “Chronological age, biological age, and individual variation in the stress response in European starlings: A follow-up study” is a rigorous examination of the relationships among telomere shortening and the glucocorticoid stress response in chronologically aging birds. By using variation in developmental conditions as the basis for expected variation in ‘biological aging’ (the rate of telomere shorting) this study is well positioned to address the correlation, if any, between telomere shortening and declines in corticosterone (CORT) response to an acute stressor. The manuscript is very well written and thoughtfully organized and the replication of work is meaningful. One minor comment is that work by Rensel and Schoech (2011) could be cited regarding consistency in CORT response (e.g., line 78, 305).

Experimental design

I have two requests for additional analysis that I think could strengthen the independent contribution of this work and the interpretation of findings.

First, I appreciated the approach of placing a smaller study in the context of similar work by including a meta analysis of all the data available. However, a direct analysis of the effects of the developmental treatments in the present study on telomere shortening was not reported outside of the metadata. I’m very curious to know the results of such an analysis and suspect it would strengthen the paper. Running and including this analysis might also clarify the effects of the treatments on ‘biological aging’ and thus bring some new insights to the last paragraph of the discussion, which struggles to explain unexpected treatment effects on the glucocorticoid response (also inferred to decline with ‘biological age’).

Second, this smaller study follows upon a prior examination of the relationship between chronological age, telomere shortening, and glucocorticoid response by directly measuring age-based reductions in corticosterone elevation in response to an acute stressor. This is an important gap to fill and the present study is designed to do just that, yet the statistical approach does not include a model with a repeated measure design (rather, two separate models are run, one at each point in chronological age). If I understand the statistics correctly, the value of this work is somewhat diminished by the omission of this test, which would have answered whether change in telomere length is related to change in corticosterone response. I see that the authors chose their current statistical approach to mirror the methods in the prior study, but isn’t the experimental design of the present work specifically aiming to contribute repeated measures?

Validity of the findings

The approach of replicating a study, even with a smaller sample size, is extremely strong and I appreciated the strategy of integrating this work with prior data. In several places the authors suggest that it was small sample size that limited their ability to replicate prior findings (e.g., line 289, 308). I appreciate the desire to avoid speculation, but it seems these results could also reflect something about the strength of the relationships among developmental conditions, ‘biological aging’, and changes in CORT response. Could the authors comments on the biological importance of these relationships and provide some ecological context such as the average life span of a European starling?

Additional comments

I have a few minor questions and comments in addition to the general comments, listed by line, below:

Line 65 In other studies and earlier in the introduction I understood ‘biological aging’ to be synonymous with telomere shortening yet this sentence seems to be relating telomere shortening and declining CORT response (which is also being proposed as a proxy of biological aging, if I understand correctly). I am simply confused as to how telomere shortening is inferred to reflect ‘biological ageing’ without lifespan data being referenced. Forgive me if I am over thinking this but perhaps it is simpler to cite existing evidence that DTL reflects lifespan?

Line 201 I would add that baseline CORT was ‘lower in chronologically older birds’.

Line 285 Here or somewhere in the paper could you include the sample sizes in Andrews
et al., 2017?

Line 322 Is there a citation that provides evidence that begging increases the rate of
telomere loss (which is what I infer from the statement that it ‘accelerate biological aging’)?

Reviewer 2 ·

Basic reporting

- Abstract is not very clear and feels choppy. After reading the method and results, the abstract makes sense. I suggest rephrasing sentences in the abstract to be clearer and flows better, particularly after “unlike the original study…..”
- Writing is not clear at some places. I pointed out where those spots are below.

Experimental design

- In the introduction or discussion, it would be helpful to lay out the rationale for using Effort in the experiment. What was the purpose of this manipulation and what was the hypothesis and prediction?
- The analysis of the change in corticosterone between 15 and 30 min was informative and adds to our understanding of adrenocortical responses in starlings.

Validity of the findings

- I support replication of previous studies which we see so rarely in our field. The authors went further than simply replicating the experiment by conducting meta-analysis and statistical simulation which strengthen the comparisons and overall conclusion which can be made.
- Effects of family (genetics) at 2 ages are interesting but I am not sure if repeatability across 2 ages can be confidentially determined because of differences in antibodies and inter-assay variation of corticosterone assays. My suggestion is to remove the repeatability results and discussion related to them. I believe removing these will not impact the importance of this paper.
- The meaning of correlation between delta TL and CORT values are not discussed in this paper. I assume this is because it has been discussed in the Andrews et al (2017) paper. However, it would be helpful to a reader who has not read the Andrews paper to have this either in the introduction or in discussion section.

Additional comments

I completed reviewing a manuscript #29791 by Gott et al. The authors aimed to replicate the study published by Andrews et al (2017) with an exception of measuring stress response at 2 ages. Unlike the previous study, this paper did not find a significant correlation between developmental telomere attrition and corticosterone parameters. When this study was analyzed with previous 2 studies in a meta-analysis, there was a significant correlation between developmental telomere attrition and the difference in corticosterone levels between time 15 and 30 minutes. Aside from major comments listed above, I have minor comments (see below). I hope my comments will be helpful in improving readability and conclusion drawn from the data.

Minor comments:
Abstract:
Line 24: In the method section, authors states that stress response was measured from 30 individual at age 2. In the abstract, it states 27 starlings. Please clarify the difference in sample size.
Lines 26-28: It is little confusing to know if this is an experimental paper or meta-analysis paper or both.
Lines 25-32: The abstract seems to highlight the shortcomings of the experiment more than what the results of the experiment shows. I suggest rewording the paragraph to describe results, what they mean, and caveats.
Introduction:
Line 43: I suggest removing “would suggest”.
Lines 70-72: I am a bit confused about this sentence. Why having quartets of siblings allow investigation of effects on baseline and peak CORT but not on the change between 15 and 30min? Please clarify.
Line 93: What do authors mean by “familial resemblance”?

Materials and Methods:
Lines 105-106: I assume n = 8 for 4 treatment groups with total of 32 birds. Please specify this.
Line 108: I suggest adding “as the following” or something similar after “under controlled conditions” so it is clear to the reader that the food and begging effort manipulation were only during day 5 and 15 post-hatch. Alternatively, the authors can state this duration close to the beginning of this paragraph.
Lines 121-123: This sentence is not clear to me. It may be me but would it be possible to simply say delta TL of each individual was compared to the mean delta TL to obtain the corrected delta TL? I am not sure what “expected regression to the mean in imperfectly correlated repeated measures” means.
Lines 128-130: Does this sentence mean that birds were separated from a group housing to an individual housing 3 to 27 days prior to blood sampling? Please clarify.
Lines 142-143: This sentence is missing something. “Blood sampling was (collected?) by puncturing an alar or……”?
Lines 157-158: I assume this was out of authors’ control but why were 2 different antibodies used in 2 years of assays?
Lines 162-163: Why are concentration for 2 control samples were expressed in %?
Lines 218-221: With different batches of assays and antibodies used, would it be possible to infer repeatability of adrenocortical responses across 2 ages?

Discussion:
Lines 275: I suggest switching “to” and “clearly” so it reads “our failure to clearly replicate”.
Line 289: What is the sample size? I noted this under method section above.
Lines 294-296: I am not sure what these 2 sentences mean. Please rephrase.
Lines 322-324: One can argue that aging HPA axis would make keep the CORT levels high rather than return to baseline quicker. Please elaborate or cite here.

---

## Round 0.2 · accepted · Accept

Thanks for responding to the reviews so thoroughly!

#